# Implementation and Strategies of Community Music Activities for Well-Being: A Scoping Review of the Literature

**DOI:** 10.3390/ijerph20032606

**Published:** 2023-01-31

**Authors:** Soo Yon Yi, Aimee Jeehae Kim

**Affiliations:** 1Department of Music Education, Graduate School of Education, Gachon University, Seongnam 1342, Republic of Korea; 2Department of Musicology and Culture, Music Therapy Major, Graduate School, Dong-A University, Busan 49315, Republic of Korea

**Keywords:** community music, well-being, scoping review, implementation

## Abstract

The benefits of community music activities for promoting well-being have been well recognized in previous literature. However, due to their wide variability and flexible approaches, a comprehensive understanding of the research and practice of community music activities for well-being promotion is sparse. The purpose of this scoping review was to synthesize published literature pertaining to community music activities for well-being promotion and identify key implementation characteristics and strategies to inform future practice and research. Studies of community music activities that investigated well-being outcomes in participants of all ages and conditions were eligible for inclusion. Through electronic database and manual searches, a total of 45 studies were identified and included in the analysis. The main findings showed that community music activities for well-being were characterized by a wide range of populations and applications, collaborative work, an emphasis on social components, and musical accomplishments. However, this variability also revealed a lack of consistent and thorough information as well as diversity in well-being conception across studies. The review offers practical recommendations for future research and practice based on the current findings.

## 1. Introduction

The field of community music has expanded rapidly over the past decades. Community music activities encompass a wide variety of forms and approaches that enable flexible application [1,2,3]. Community refers to a “group of people with diverse characteristics who are linked by social ties, share common perspectives, and engage in joint action in geographical locations or settings” [4] (p. 1936). Community music comprises a vast array of variations and approaches in its conception and practice that depend on the emphasis placed on shared perspectives and values as well as the music activity and experiences [2,3], which can make it difficult to reach a consensus on a definition. Despite this complexity, previous literature suggests that the central aspect of community music activities is the act of actively engaging in shared music making with common interests that people can enjoy [1,5]. In school settings, community music activities have been provided to students as means of providing satisfying music experiences that are challenging to provide in top-down class structure [6,7]; in the social context, community music activities have been applied to a broader population as a social intervention to serve various needs of individuals and society [5,6,8,9].

Particularly, benefits on well-being have been considered key to community music activities. The significance of well-being has gained attention with the rise of positive psychology, which focuses on positive functioning, life satisfaction and happiness for mental health [10,11] rather than focusing on mental illness [12]. Research on well-being has focused on two approaches for well-being: hedonic and eudemonic well-being. The hedonic approach to well-being emphasizes positive emotional states, such as happiness and pleasure, while the eudemonic approach emphasizes full functioning of a person, including aspects of meaning, purpose, and self-actualization [10,11,13]. Additionally, the social aspect of well-being, including social integration, social contribution, social coherence, social actualization, and social acceptance, is identified as a key factor that impacts mental well-being [14]. Hence, consideration of the emotional, psychological, and social dimensions of well-being is crucial in well-being research and intervention.

Community music has been recognized as having a positive impact on well-being across various population. In particular, research has shown that community music activities can benefit all emotional, psychological, and social dimensions of well-being [15,16]. Individuals who regularly participate in choral singing have reported that their participation in the choir contributes to their well-being through increased social capital and positive emotions as well as providing meaning and purpose in their lives [17]. Additionally, participating in group singing has been reported to improve well-being in adults with mental health conditions by providing emotional and social benefits and reinforcing self-efficacy [16]. Specifically, active music engagement is considered a crucial component of community music activities [1,18]. A recent meta-ethnography involving over 2000 participants suggested that the multifaceted process of participatory music engagement supports mental well-being for various individuals by allowing them to engage in specific processes tailored to their individual needs and circumstances [19].

Despite the growing research and practice of community music for well-being, there is a lack of information on its organization and implementation. In current research, community music implies to a wide range of music activities and community contexts [2,3,20]. While this diversity reflects the wide applicability of community music, it also creates ambiguity in providing guidelines for planning and implementing community music activities in practice and research. Specifically, contextual characteristics, such as community needs, organizational structure, and resources [3,21], as well as specific implementation strategies for music activities that are tailored to the community [3], are suggested as an integral part of the desired outcome. The aim of this review is to synthesize published literature pertaining to community music for well-being and identify the key implementation characteristics and strategies to inform future practice and research.

## 2. Materials and Methods

To investigate the extensive array of practice and research of community music activities and identify comprehensive findings and gaps, we chose to conduct a scoping review to identify the key characteristics of community music activities for promoting well-being in literature, rather than a systematic review that focuses mainly on formal assessments of studies with a relatively narrower scope. We conducted this review using the five-stage framework by Arksey and O’Malley [22] and the PRISMA-ScR (Preferred Reporting Items for Systematic reviews and Meta-Analyses extension for Scoping Reviews) checklist [23] as methodological guidelines.

### 2.1. Stage 1: Identifying the Research Question

According to the study objective, we pose the following research questions:What are the key characteristics of existing studies of community music activities for well-being?What are the implementation characteristics and strategies of community music activities for well-being?What are the benefits of community music activities on promoting wellbeing in terms of study outcomes?

### 2.2. Stage 2: Identifying Relevant Studies

To identify the key articles, we developed inclusion criteria based on the guideline suggested by the JBI Manual for Evidence Synthesis [24] (Table 1). A comprehensive search strategy was applied from 6 July 2022 to 13 July 2022 and used multiple sources [22], including electronic database search, manual search of key journals, and reference lists from inception date to August 2022. First, an electronic database search was carried out in the following databases: Scopus; Web of Science; PsychINFO; ProQuest; PubMed; Cochrane library; and ERIC. We used keywords related to community music activities and well-being as search terms, including “community music”, “singing”, “ensemble”, “orchestra”, “choir”, “well-being”, and “mental health” in varying combinations using Boolean operations AND and OR to identify relevant literature (Appendix A). Second, we conducted a manual search of relevant journals including *Psychology of Music*, *Musicae Scientiae*, *Journal of Music Therapy*, and *Arts and Health*, and reference lists of identified literature. 

### 2.3. Stage 3: Study Selection

After removing duplicates, two authors independently screened titles and abstracts for eligibility. Articles with insufficient information to confirm inclusion were included in the full-text review. The remaining full-text articles were assessed independently by two authors for eligibility by checking the inclusion and exclusion criteria and then reviewed by both authors until inclusion agreement was reached.

### 2.4. Stage 4: Charting the Data

Key information was extracted using a data extraction form created by authors based on the research questions identified in stage 1. For the first research question regarding key characteristics of selected publications, categories included authors, year of publication, study location, study design, participant, and community characteristics. We extracted community characteristics to identify the target community differentiated from individual characteristics by descriptions of authors. For records in which community characteristics were not specified, we summarized the overarching shared quality of the participants that enabled community formation. These included characteristics of community music activities, including type of project and music activity, facilitator, and delivery schedule and duration. As we sought to comprehensively review the existing literature, both ongoing and planned projects were included. Ongoing projects were community music activities that were not planned for research purposes and that are continually being conducted. Planned projects included activities planned for a limited time and had a schedule comparable to interventions. 

Implementation characteristics and strategies were summarized in two areas: (1) contextual components of community and (2) components of music activities. Contextual components of community included categories of activity goals based on community needs and organization partnership as well as institutional and operational partnership and resources. The categories were developed based on results of a systematic review on community-based organization [21]. Components of music activity included music strategies related to the use of music, activity strategies regarding structures and approaches, and participant involvement, used to summarize the active involvement of participants in the activity. Finally, the benefits of community music activities on promoting well-being were extracted by summarizing and synthesizing the study results for both quantitative and qualitative well-being outcomes. Critical appraisal on included publications was not conducted as scoping reviews focus on developing a comprehensive overview of literature regardless of methodological quality, unlike systematic reviews [25].

### 2.5. Stage 5: Collating, Summarizing and Reporting Results

Initially, two authors independently extracted data from the included publications, then refined and reviewed categories through multiple discussions to achieve the final papers included in the review. When necessary, information from publications identified as duplicate or multiple publication by trial numbers was collated. 

## 3. Results

### 3.1. Key Characteristics of Studies

#### 3.1.1. Publication Characteristics

A total of 46 publications met inclusion criteria. However, two of these publications were identified as multiple publications of the same intervention by trial registration number [26,27]. In accordance with the deduplication guidelines provided by The Cochrane Handbook [28], we collated the data from these two studies into one study unit. Therefore, a total of 45 studies were included in the analysis. The study selection process and details are depicted in Figure 1. 

#### 3.1.2. Participant Characteristics

The demographic reporting among the included studies varied widely. Sample sizes in the studies ranged from 5 to 390 participants. Specifically, between 5 and 47 participants were included in qualitative studies, between 20 and 342 in mixed methods studies, and between 41 and 382 in quantitative studies. Participants’ ages ranged widely, from 3 to 95 years. However, some studies did not provide age information (*n* = 5; 11%) or did not provide a maximum age (*n* = 5; 11%). A few studies included mixed age groups, such as adolescents to middle-aged adults [29,30], and young adults to older adults [31,32,33]. 

Participants also varied in population. We summarized both individual characteristics and community characteristics to investigate the communal aspects of the participants. In terms of individual characteristics, 24 studies (53%) involved non-clinical population and 21 studies (47%) involved populations with clinical conditions or experience. Of the studies involving non-clinical populations, the group most commonly studied were older adults (*n* = 8). Three studies included school children and adolescents [34,35,36], and studies that included community music activities for a broader population including both children and adults or open groups were also reported [30,37,38,39,40]. Studies on populations with specific needs were reported, including patients’ caregivers and bereaved partners [41,42,43], parents and mothers [44,45], and immigrants, refugees and asylum seekers [46,47].

A total of 21 studies involved participants who either currently have or have had clinical conditions but are not hospitalized. Ten studies included participants with mental health conditions, needs, or previous experiences, and eight studies included people with dementia (PwD) in their population. Studies also reported on populations of individuals with learning disabilities [29], chronic obstructive pulmonary disease [48], chronic disease [49], and prostate cancer [50]. Interestingly, nine studies that reported on a clinical population included caregivers, family, and supporters; seven studies specifically targeted groups of caregiver-PwD dyads [33,51,52,53,54,55] and two studies targeted people with mental conditions [32,56]. One study also included both individuals from the general population and those with mental health conditions [57]. Descriptions of participant characteristics are summarized in Appendix B, Table A1.

#### 3.1.3. Community Characteristics

Summary of the community characteristics in the included studies indicated that socio-economic factors, regions, and affiliation were the main determinants of the formation of the community group. In 15 studies (33%), the participants were socially disadvantaged; this included individuals from low socio-economic areas [34,35,36,40,58], those who lacked social support because of chronic health problems or older age [26,27,59,60], and those who were facing adversity because of social circumstances or cultural diversity [39,44,46,47,61,62,63]. There were also studies where participants shared the same cultural backgrounds [39,46,64]. A total of 13 studies (28%) included participants who lived in the same area and shared common denominators, such as health conditions, non-health related issues, or interests. Community groups formed through affiliated institutions such as schools, community centers, and healthcare facilities were reported in 10 studies (22%). It is worth noting that the community groups of most studies included more than one of the above-mentioned community characteristics. Summaries are shown in Appendix B, Table A1.

#### 3.1.4. Music Activity Characteristics

Out of the 45 studies included, 24 (53%) were planned projects and 21 (47%) were ongoing projects. The most frequently used music activity in the studies was group singing (*n* = 28; 62%), followed by music performance (*n* = 16; 36%), choir (*n* = 13; 29%), creative music making, ensembles, music recording (*n* = 7; 16%), music learning (*n* = 5; 11%), and music and dance (*n* = 3; 7%). Orchestra or band activities and instrument playing were categorized as ensembles. While some studies (*n* = 21; 47%) only used one type of music activity, many of the studies (*n* = 23; 53%) used a combination of types. Specifically, 12 studies used two types and 11 studies used more than three types, with music performance being the most frequently combined activity type in 16 studies (36%).

The majority of the studies were identified as having more than one facilitator, with professional musicians being the most common facilitators (*n* = 25; 56%), including professional choir leaders and directors, followed by non-musicians (*n* = 8; 18%), trained facilitators (*n* = 7; 16%), music teachers (*n* = 6; 13%), and music therapists (*n* = 4; 19%). Five studies (11%) did not provide facilitator information. Multidisciplinary cooperation was reported in seven studies (16%), such as a music teacher, music therapist, and social workers working as a team [65], or musicians working with volunteers or staff members [53,66].

Only 34 articles (76%) among the included studies provided information on session delivery schedule, the descriptions of which varied widely. The number of sessions ranged from 3 to 44 sessions among the 28 studies (62%) that provided this information. In terms of delivery schedules, 26 studies (58%) reported that they provided weekly sessions, and in 16 studies (36%), duration per session ranged from 1 to 3.5 h. Details are shown in Appendix B, Table A1.

### 3.2. Implementation Characteristics and Strategies

#### 3.2.1. Contextual Components

Goals of community music activities were summarized in 39 studies (87%). We sought to identify the activity goals reported in studies that were distinguished from research objectives by identifying the need to provide the community music program. However, we note that descriptions were inconsistent in majority of the studies. Thus, we summarized the community needs when described by authors, and otherwise summarized the program goals, when provided. Summaries are shown in Appendix B, Table A2. 

Results showed that most studies indicated more than one goal area. Well-being promotion was reported as an activity goal in 23 studies (51%), including improving quality of life. Goals related to social support and social well-being were shown in 17 studies (38%), such as supporting social inclusion and social engagement [30,31,38,54,65,67], promoting relationship and social interaction [33,37,51,52,54], and supporting community [34,56,64]. Health improvement, including both physical and mental health, was reported in 14 studies (31%). A total of nine studies (20%) indicated their goals as providing music activities per se because the activities themselves were positive [36,41,51,61,62] and provided a meaningful and enriching [44,47,60] experience. Psychological goals were reported in seven studies (16%), including supporting self-dependence [61,62], confidence and self-efficacy [35,36,42] and resilience and empowerment [49,57]. 

Organization partnerships and stakeholders were summarized in 41 studies (91%). We summarized institutional partnership as collaborations between institutions for initiating and planning the community music activity; we defined operational partnerships as referring to co-work in implementation and operation. A total of 29 studies (63%) reported institutional partnerships. Public and private social and health care organizations and community centers had the highest rate of involvement (20 studies; 44%). Academic institutions, such as universities and research institutions, and music organizations, including professional orchestras, choir groups, and community music groups, were involved in nine studies (20%). Partnerships with governmental institutions (*n* = 8; 18%) and charity organizations (*n* = 5; 11%) were also reported. The number of institutions involved in the institutional cooperation varied among studies. Operational partnerships were reported in 23 studies (51%) that included assistance and co-work between volunteers and various professionals in facilitating sessions, recruiting participants, and coordinating and organizing events (see Appendix B, Table A2).

Resources reported in the studies mainly included funds and venues. Funding resources were reported in 14 studies (31%). Funds from community organizations (*n* = 7; 16%) and governmental funds (*n* = 6; 13%), including research funds (*n* = 4; 10%), were reported most often; five studies (11%) also reported resources provided by charity groups and donations. Multiple fund resources were reported in four studies (9%), included membership fees and fundraising [39]. Additionally, 13 studies (24%) reported information on the venue for sessions and performances.

#### 3.2.2. Music Activity Strategies

Music activity strategies were analyzed in three categories: music strategies, activity strategies, and participant involvement. Music strategies included reports of strategies specific to music selection, delivery, or performance, while activity strategies concerned strategies for session structures or contents. Participant involvement was analyzed based on activities or suggestions made by the participants to identify activities that were participant-led. Details are shown in Appendix B, Table A2.

Reports on music strategies were documented in 36 studies (80%). Music selection criteria were specified in 27 studies (60%). Familiar music which was preferred or requested by participants was used in 15 studies (33%). A total of 15 studies (33%) specified music selection criteria based on a specific functional purpose. For example, studies selected music and genres that reflected the participants’ cultural background [27,39,45,46,55,68], with two studies specifically using songs that had language which corresponded with the culture [45,55]. Other strategies included using new songs in combination with familiar songs to facilitate cognitive stimulation in the older population [41,42,48,51,64,66] and selecting songs linked to specific memories [40,51,54,69]. Only five studies (11%) provided the genre of the music repertoire; various genres, including popular music, were mentioned. A total of 10 studies (22%) indicated who selected the music; 7 studies (16%) reported using songs that were suggested, requested, or agreed to by the participants, and 3 studies (7%) used facilitator chosen songs.

Music strategies regarding the use of instruments were reported in 15 studies (33%). A number of studies described providing instruments [29,32,51,53,66,67,70] and music technology [58] to offer additional musical engagement opportunities. Results also showed strategies using instruments for live accompaniment [43,48,51,54,60,67,68,70] and selecting instruments based on type of songs [43,71] and ethnic diversity [39].

Among the 18 studies (40%) that specified activity strategies, 14 studies (31%) used strategies to minimize difficulties that could prevent participants from engaging in music activities, such as lyric sheets and song books [43,45,48,59,60,67], learning songs by ear [31,41,71], and tailoring the music piece [41,54,60,64] or music activities [29,37,53] to the participants’ musical ability. Interestingly, few studies indicated strategies that structured the music piece according to participants’ psychological needs, such as singing in unison to avoid feeling anxious [43] or providing harmony parts for social support [54]. Strategies to maximize music engagements were reported in three studies, which provided a wide range of music activities to provide more musical opportunities [65,70,72]. In terms of session structure, 13 studies (29%) specified providing vocal or/and physical warm-ups and 10 studies (22%) reported integrating social components, such as providing time for socializing during tea breaks or refreshments.

Participant involvements in music activities were analyzed by summarizing the active roles and participant-led elements within the music activity. Participants actively taking part in various forms of performances was the most reported form of active involvement in 15 studies (33%), followed by involvement in music and activity selection which was reported in 12 studies (12%); this included suggesting songs and selecting the instrument or type of activity to participate. Other active involvement included participation in the creative composition process [37,40,44]; some studies included music recordings, practicing or participating in music programs at home [49,54], and taking part in preparing for activities [55].

### 3.3. Study Outcomes

#### 3.3.1. Quantitative Outcomes

Quantitative outcomes were summarized in 21 quantitative and mixed methods studies (47%). The well-being outcomes and measurements in each of the included studies varied widely. Outcome domains included well-being, emotion and psychological, QoL, life satisfaction, physical health, behavior, daily living, cognition, and social outcomes. Most studies used multiple measurements to examine the effect of participating in community music activities on well-being. The overall results of quantitative outcomes demonstrated inconsistent outcomes. Detailed information on measurements and quantitative results are summarized in Appendix B, Table A3.

Well-being measures were used in six studies, including six outcome measures of Warwick Edinburgh Mental Wellbeing Scale (WEMWBS, *n* = 3) [31,41,42], short version of the Warwick Edinburgh Mental Well-being Scale (SWEMWBS, *n* = 1) [45], social well-being (ScWB, *n* = 1) [26], and the Stirling Children’s Well-being Scale (SCWBS, *n* = 1) [36]. Significant results on well-being measures were reported in three studies on WEMWBS [32,41,42]. A total of 14 studies reported outcomes on emotion and psychological measures, including 27 measurement tools, such as: the Patient Health Questionnaire scale (PHQ, *n* = 3) [30,54,64]; GDS [51,59]; HADS [41,42]; Depression Anxiety Stress Scale (DASS) [27,54]; UCLA Loneliness Scale Version [26,59]; Rosenberg Self-Esteem Scale (RSS) [26,41]; rating on mood energy, concentration and stress level [55,59] (*n* = 2, respectively); Depression Scale (GAD) [30]; Rating Anxiety in Dementia Scale (RAID) [54]; General Self-Efficacy Scale (GSE) [41]; Positive and Negative Affect (PANAS) [25]; Clinical Outcomes in Routine Evaluation questionnaire 10 (CORE-10) [32]; Clinical Outcomes in Routine Evaluation questionnaires (CORE-OM) [31]; Apathy Evaluation Scale (AES) [54]; Cohen-Mansfield Agitation Inventory-Short Form (CMAI-SF) [54]; Control, Autonomy, Self-realization, and leisure scale (CASP) [72]; Basic Psychological Needs Scale (BPNS) [72]; Strengths and Difficulties Questionnaire (SDQ) [36]; and the Positive Affect and Apathy, and Positive Aspects of Caregiving Questionnaire (PACQ) [54] (*n* = 1, respectively). Eight studies reported significant results on emotion and psychological outcomes were reported in the outcomes among 11 measures. However, outcome results were inconsistent.

QoL measurements were used in five studies, including Health-Related Quality of Life and Well-Being (York-SF 12, *n* = 2) [49,69], World Health Organization Quality of Life: Brief Version (WHOQOL, *n* = 1) [26], Dementia Quality of Life (Dem-Qol, *n* = 1) [51], and Quality of Life-Alzheimer’s Disease (QoL-AD, *n* = 1) [55]. Significant results on QoL outcomes were reported in three studies, including both studies that used the York-SF 12 scale and one study that used the social relationship subscale from WHOQOL. Life-satisfaction measurements were used in four studies, including five measurement scales of Flourishing and Satisfaction with Life scales (SWLS, *n* = 3) [26,30,54], Life Satisfaction (ONS, *n* = 1) [58], and Flourishing Scale (FS, *n* = 1) [54]. Significant results regarding life-satisfaction outcomes were demonstrated in only one study using the ONS scale.

Physical health outcomes were measured in six studies that used the Health-Related Quality of Life (EQ-5D, *n* = 2) [49,69], Medical Outcomes Study Short-Form Health Survey (SF-36, *n* = 1) [59], Medical Outcomes Study Questionnaire Short Form 36 Health Survey (MOS 36-SF, *n* = 1) [30], unipedal stance test (*n* = 1) [26], and standing balance measure (*n* = 1) [64]. Significant results were reported in two studies that used the EQ-5D and MOS 36-SF scales. Studies including outcomes on cognitive functioning (*n* = 3) [51,55,64], social functioning(*n* = 2) [26,30], and behavior (*n* = 1) [51] demonstrated no significant effects (Appendix B, Table A3).

#### 3.3.2. Qualitative Outcomes

All studies that provided qualitative outcomes of participants’ well-being experiences were included for synthesis. Two authors independently extracted subthemes from qualitative outcomes relevant to supporting well-being by thoroughly reading the quotes, description, and discussion by the study authors. All subthemes were re-examined and discussed between authors, resulting in the generation of well-being themes that captured the meaning of subthemes. The qualitative outcomes of 34 studies were included in the analysis. Study outcomes that did not involve qualitative methodologies [17,45,55] were excluded from the final analysis. The synthesis resulted in 52 subthemes, categorized in 13 well-being themes and 5 well-being domains of physical, cognitive, emotional, social, and psychological well-being. Details are shown in Table 2.

For physical well-being, 12 studies reported physical benefits for those who participated in the community music activity, which included subthemes of general physical benefits [38,59,60,62,72], physical benefits regarding health conditions related to diagnosis [40,50,57,66,68,71], and vitality [50,53,60]. Emotional well-being benefits were reported in 33 studies, including experiences of positive emotions and coping in terms of dealing with negative or difficult emotions.

For positive emotions, experiences of enjoyment, happiness, and pleasure [29,32,33,34,35,36,38,39,51,52,53,57,59,67,68,70,71,72] were frequently reported, along with heightened arousal [63] and relaxation [29,32,60,65]. Studies reporting benefits on coping indicated that the community music activity improved participants’ ability to cope with stress and negative emotions [40,47,50,58,62,63,67,68,70,72], provided opportunities to release emotions [40,56,63,65], and allowed for self or emotional expression [29,40,56,57,60,65,72].

Social well-being benefits were reported in 33 studies including well-being themes of positive relationship (*n* = 24), social belonging (*n* = 25), reciprocal support (*n* = 23), feelings of contribution (*n* = 7), and engagement (*n* = 19). Experiences of positive relationships through community activities were indicated by participants building new relationships or supporting existing relationships [26,29,32,33,34,35,36,39,40,43,50,52,57,59,65,67,70,71], and having positive interactions with others [26,29,33,39,56,60,61,63,66,67]. Support for the sense of social belonging was reported by participants’ experiences of feeling connected [38,44,47,50,52,58,60,61,62] and belonging [32,33,35,39,40,43,46,47,50,51,52,53,57,63,65,67,72] to the group, and feeling a sense of cohesion [39,46,47,50,56,57,65,66]. Studies also showed reports of empathizing with others in the group [52,56] and feeling culturally understood [46] within the group.

Reciprocal support through community music activities was indicated in 23 studies that described how gathering regularly provided a sense of support and reciprocity on its own. Furthermore, participants’ feeling of making a contribution to the group and others (*n* = 7) [33,38,50,53,62,63,72] and the benefits of being musically and socially engaged (*n* = 19) were shown as social well-being benefits.

Finally, 30 studies included participant experiences of benefits in terms of community music activity on psychological well-being, such as personal growth (*n* = 11), self-acceptance (*n* = 19), sense of purpose (*n* = 21), and sense of accomplishment (*n* = 25). Statements of personal growth also included participants gaining perspective [40,44,65], having spiritual experiences [40,46,50], feeling rejuvenated [33,62,72], and experiencing increased resilience [32,47] by participating in the community music activity. Studies also reported that participation increased participants’ self-acceptance and positive attitudes about themselves through being more self-aware [56,61,65] and more aware of their own identity and individuality [35,43,46,47,50,52,57,58,62,70]; they also experienced increased personal validation [35,46,47,48,53,57,60,61,62,63,68], including increased self-esteem and self-worth. Participants also reported benefits of feeing a sense of accomplishment that gave confidence and pride. Description of participants having a sense of purpose also included statements of being more enthusiastic and motivated [26,36,38,48,51,59,62,70,72] and having found a sense of meaning [44,52,53,72]. Summaries of well-being themes among studies are shown in Table 2.

**Table 2 ijerph-20-02606-t002:** Summary of qualitative results.

Well-Being Theme	Description	Studies
**Physical well-being**		
Physical benefits (12)	Physical benefits related to general health or health condition and diagnosis	[38,40,50,53,55,57,60,62,66,68,71,72]
**Cognitive well-being**		
Mental work (25)	Stimulating cognitive processes, such as focusing, learning, and creativity	[29,33,35,38,40,47,48,50,51,52,53,56,59,60,61,62,63,65,66,67,68,70,71,72]
**Emotional well-being**		
Positive emotion (31)	Feeling positive emotions, such as joy, happiness, and relaxation	[27,29,32,33,34,35,36,38,39,40,44,46,47,48,50,51,52,53,57,59,60,61,62,63,65,66,67,68,70,71,72]
Coping (19)	Coping with negative emotions and stress by means of music activities, such as expressing oneself	[29,32,40,47,50,56,57,58,60,61,62,63,65,67,68,70,72]
**Social well-being**		
Positive relationships (24)	Building new relationships and supporting existing relationships	[26,29,32,33,34,35,36,39,40,43,50,52,57,60,61,62,63,65,66,67,70,71]
Social belonging (25)	Feeling connected, accepted, and included in the group	[32,33,35,38,39,40,43,44,46,47,50,51,52,53,56,57,58,60,61,62,63,65,66,67,72]
Reciprocal support (23)	Feeling supported by regular gatherings of group	[33,35,36,39,40,43,47,48,50,52,53,56,57,58,59,60,62,63,65,66,67,70,71]
Feeling of contribution (7)	Feeling that one has contributed to the group and others	[33,38,50,53,62,63,72]
Engagement (19)	Being musically or socially engaged	[26,29,33,34,36,38,39,44,47,52,53,56,57,58,61,62,63,70,72]
**Psychological well-being**		
Personal growth (11)	Experiencing sense of personal growth, such as spiritual experiences, gaining perspective and feeling rejuvenated and resilient	[32,33,40,47,50,60,62,65,68]
Self-acceptance (19)	Being aware of and accepting oneself with positive attitudes	[35,43,47,50,52,53,56,57,58,60,61,62,63,65,66,67,68,70,73]
Sense of purpose (21)	Having a sense of purpose and meaning in life	[26,29,32,33,36,38,39,43,48,50,51,52,53,59,62,65,68,70,72,74]
Sense of accomplishment (25)	Feeling a sense of accomplishment that gives confidence and pride	[26,29,32,33,34,35,36,39,40,48,50,51,53,57,58,60,61,62,63,65,68,71,72]

## 4. Discussion

The aim of this scoping review was to comprehensively synthesize and identify key characteristics of research and implementation in publications of community music activities to promote well-being, and further, to identify gaps in current research and key knowledges of implementation strategies that can inform future research and practice. For this purpose, first, we summarized study characteristics in terms of publication, population, and music activity characteristics. Second, we identified the implementation characteristics regarding the contextual components and music implementation strategies of the community music activity. Finally, we summarized the quantitative and qualitative results of included publications to address the well-being benefits of community music activities. 

### 4.1. Study Characteristics

Findings showed that publication frequency has increased in the past 20 years, particularly in the past 10 years, indicating a growing interest in field. However, the results showed that the studies were predominantly carried out in the UK and Australia, suggesting an as yet circumscribed implementation and interest in other regions, which has been also stated in previous literature [6]. In terms of study design, the majority of the studies employed qualitative methodologies, including mixed methods studies. This shows a focus on subjective experiences and processes of changes in community music activities rather than well-being effects, which captures the multifaceted, complex, and contextual nature of community music [5]. 

This review found a wide range and diversity of participants in the included studies; this was in fact one of the participants’ key characteristics. Study participants ranged from non-clinical to clinical populations; children to older adults; and people with specific needs to the general public. These results reflect the distinct characteristic of community music as a social intervention for various needs of individuals and society; those needs encompasses education, healthcare, and welfare, as reported in the previous literature [6,8,9]. The review showed that both clinical and non-clinical populations were included as participants. Among non-clinical populations, studies on older adults were most commonly reported, emphasizing the benefits of participating in music activities for healthy aging [75]. Participants with mental health conditions and dementia were most often considered in clinical population studies, implying the need for additional psychological and social support in that population. Inclusion of family members was reported in a few studies that examined the importance of recognizing caregiver distress and caregiver-patient relationship; this was also reported in previous studies [76,77]. The findings of community music activities with clinical population share contexts with community music therapy that focuses on the well-being, social, and health benefits of socially engaged music making [78,79].

Another notable characteristic was reports on the inclusion of mixed age groups, i.e., of including people from young to older ages within the community music group [29,30,31,32,33]. Comparable findings were identified in previous literature [80] indicating that community music groups can be formed through musical and social motivations. This suggests that the motivation to experience music in a group can play a role in forming a community music group [2]. The results imply that the focus on such motivational factors can be a distinctive feature of community music activity group which is distinguished from other music-based interventions.

Socio-economic factors, regions, and affiliation were primarily considered for community music group formation in the included studies. Community music groups representing socioeconomically and culturally diverse groups have often been reported in previous literature [2,80], however, the current review showed that institutions also played a significant role in forming a community music group, such as schools, community centers, and healthcare facilities. Groups and gatherings were organized through institutions to provide further support based on the needs of the members. The findings indicate that engaging such institutions and facilities may support the initiation of music groups for community members with the specific needs characterized by the type of institution.

Group singing was found to be the most applied music activity in the included studies, followed by performance and choir. Use of activities such as creative music making, music ensembles, music recording, music learning, and music and dance were also noted in studies. Group singing has advantages in its accessibility for participants, as it does not necessarily require pre-skills and costly equipment but is also enjoyable. Extensive evidence on the social and psychological benefits of group singing [16,20] may also support the use of group singing for community music activities that promote well-being. Findings showed that a substantial number of studies used more than one kind of music activity, and that performance was most often used in conjunction with other activity types. Previous literature has also stated the benefits of performance on well-being, as it promotes social bonding by working together towards a common musical accomplishment [81]. The benefits of performance, including providing meaning and a sense of accomplishment, has been also identified in this review’s results on qualitative outcomes. In terms of activity facilitators, a wide variety of professions were reported, with professional musicians being the most common. Other professions included non-musicians, music teachers, and music therapists. The findings reflect the wide range of application among disciplinary fields [6], which was also evident in frequent reports of multidisciplinary cooperation by facilitation.

### 4.2. Implementation Characteristics and Strategies

Findings showed that most studies indicated more than one activity goal based on community needs. While the majority of the studies specified well-being promotion as the community goal, the included studies considered other goals related to social support, health improvement, and psychological domains. The effect of music activities on health and well-being is well documented in previous studies, suggesting rich evidence on physical, emotional, cognitive, social, and psychological effects, including self-esteem and identity [82]. The results imply that the applicability of music activities may serve the extensive and specific needs of the community. It is also noted that a number of studies indicated community goals for providing music activities in and of themselves, and that such findings can be supported by previous reports that engaging in music with others can positively affect subjective well-being [83] by contributing to living a flourishing life [81]. However, the review found that descriptions of community needs were inconsistently reported in the included studies; while most studies provided study objectives, several studies failed to provide information that specified the needs of the community. The previous literature stresses that community activities must serve the needs of the target community [21]. Moreover, it has been pointed out that the group participating in the community activity functions as an enabling context [84], thus the social aspects or social well-being related to the needs of target community should be clarified along with the individual needs.

Organizational partnerships were reported in most of included studies. Findings showed that partnerships were carried out on two levels: institutional and operational. Institutional partnerships were reported in 63% of the included studies; in these studies, the community music project was initiated and planned through cooperation of institutions, including social and health organizations, community centers, academic, governmental institutions, and charity organizations. Previous literature emphasizes the role of community organization partners in maintaining community music activities, specifically with regards to recruitment and retention of participants and funding initiatives [80,85]. This review confirms those results. In particular, the active involvement of staff and leaders are suggested to be crucial in implementation [3,80], which is shown in findings regarding operational partnerships; in these partnerships, various personnel are involved in facilitating, recruiting, coordinating, and organizing sessions. This type of partnership was reported in 51% of the included studies.

Community music activities require a great deal of non-human resources; this can include funding, instruments, spaces, and so on [3]. Financial resources in particular are essential in implementing community music activities [21,85]. Overall, the resources reported in the included studies mainly focused on funding and venues. Community and charity organizations, along with government and research funds, were identified as prominent sources if funding in this review and were reported in 30% of the included studies. As mentioned earlier, financial resources are crucial in sustaining and disseminating community music activities. For instance, efforts of UK and Australian governments for funding community music projects have been well documented [6,85], which may explain why the majority of studies included in this review come from those countries. The findings support previous arguments that government policy and funding for supporting community music initiatives are indispensable for developing community music research and practice.

Findings showed that music selection strategies were most often reported in the reviewed studies. Music that was familiar, preferred, or requested by participants was used in 33% of the studies, while 33% specified the music selection based on the needs and backgrounds of participants, including their cultural background and clinical needs. There were also studies that only reported the genre of music. However, it is important to note that a large number of studies did not provide information on the reason for music selection. Music strategies regarding use of instruments were also reported to encourage additional music engagement opportunities.

Activity strategies reported in 40% of the included studies discussed strategies related to music delivery and session structure. Music delivery strategies were used to minimize difficulties that prevented engaging in music by using additional materials or tailoring the music or music delivery according to participants’ needs. Strategies to maximize participants’ music engagement were also reported and included providing wide range of musical opportunities. Activity strategies regarding sessions included providing warm-up sessions and time for socializing. Regarding participant-led elements, summaries of participant involvement showed that performance was the most frequently reported form of active involvement, followed by music and activity selection by participants. Notably, the strategies addressed appear to facilitate participant motivation, engagement, and social bonding, which is regarded as the central aspect of community music activities [3,5].

The existing literature suggests that community music activities vary in type by whether the music making or other social or health related benefits are the primary purpose of the activity [2]. Our review revealed that publications of community music activities for well-being promotion included both types. Either way, the delivery and strategies of music activities are essential to achieve the goal. Unfortunately, by including a wide range of research on community music activities, the review showed that descriptions of music delivery or intervention varied widely. In terms of music-based intervention, systematic, protocolized, and detailed procedures are emphasized for interpretation and implementation to practice [86]. Some might argue that the complexity and diversity of community music practice cannot be translated into rigorous empirical tradition. However, research efforts to provide sufficient information regarding the music delivery process are necessary to inform practice and further expand the applicability [80].

### 4.3. Study Outcomes

The summary of quantitative outcome revealed that a wide range of measurements were used in the studies, including measurements of well-being, emotional and psychological state, QoL, life satisfaction, health, behavior, cognition, and social factors. Previous systematic reviews of music activities showed similar results regarding the inconsistency of well-being measurements in existing studies [20]. Findings may imply that the well-being aspects of community music activities embody a wide spectrum of well-being, but that a discrepancy also exists in conceptualizing well-being. Overall, quantitative outcomes across all measurements demonstrated inconsistent results. Despite some promising results, findings were inconclusive across all measurement domains, which is in line with previous comparable studies [20,60].

The inconclusive results and heterogeneity of well-being measurements may result from the multidimensional construct and aspects of well-being. Specifically, Luhmann, Krasko, and Terwiel [87] suggest that well-being has both structural and temporal facets: short-term/long-term and affective/cognitive. The authors suggest that short-term affective and cognitive well-being reflect state affect and state life satisfaction, respectively, while long-term affective and cognitive well-being relates to trait affect and life satisfaction, and that these facets can be linked to hedonic and eudemonic well-being. The redundancy in well-being concepts has been also argued. Conceptions such subjective well-being, QoL, and happiness are shown to have large overlaps [88], while questions are raised about some well-being measurements that overlook the broader sense of social and community well-being as well as other dimensions [89]. Possibly, some well-being measurements in studies may have not been in accordance with the intrinsic well-being aspects of the community music activity. Thus, prospective studies should deliberately designate measures that are true to the nature of community music activities by considering its impact on different dimensional, structural, and temporal aspects of well-being [89].

Synthesis of qualitative outcomes revealed that participation in community music activities provides benefits in various well-being aspects, including physical, cognitive, emotional, social, and psychological domains. The results showed that all studies reported benefits from multiple domains. Reports of well-being benefits in studies included generic features, such as social support, interaction, reciprocity, engagement, emotion management, and accomplishment, but also included specific features according to participants’ needs; for example, reports include physical benefits in clinical populations or relationship support in caregiver and clinical populations. Perkins et al. [19] reported comparable results on emotional, social, and psychological benefits in their meta-ethnography on participatory music engagements. The authors indicated that various well-being aspects of participatory music engagements support mental well-being through multiple pathways according to individual needs and context. Overall, the well-being themes derived from the review may also provide information on implementation strategies that can facilitate well-being by reinforcing well-being aspects in implementation specific to target population and community.

### 4.4. Recommendations

Our review investigated the research and implementation characteristics of community music activities for well-being promotion. The review revealed a wide variety and diversity among studies. Furthermore, the review showed that community music activities encompass various elements and components, regarding the community site, population, goal, and type of music activity. In this aspect, we offer recommendations for future research and practice of community music activities for well-being.

The current body of research showed limitation in its lack of consistent and sufficient documentation of key study information. We recommend providing sufficient information regarding (1) community and population characteristics; (2) specific details of the community music program; and (3) the specificity of selected outcome regarding the community music program. The review revealed that the social element and community characteristic is a crucial element to community music activities. Future research may provide sufficient information on each of the community characteristics; this could include the shared aspect and needs of the community as well as the rationale for the needs of the community music program and contextual factors. Future research should also provide full descriptions of the study population. As this review revealed the wide variety of population as a unique quality of community music activities, we recommend considering this point as a rationale describing population characteristics.

To report on the music program, reporting the following program details is recommended: (1) information on music program components and procedure, which may include program content, schedule, and human and non-human resources; and (2) strategies and rationale for music and activity selection, which may include participant involvement, interactive or social components, and strategies specific to the community and population. As our review revealed the inconsistency and heterogeneity of quantitative outcome results in the included studies, we recommend considering the well-being aspects of the program and considerately select the outcome regarding its specificity and relevance to the program. Especially, the synthesis of qualitative outcomes in this review may provide useful information for outcome selection.

For future practice and implementation, the findings indicate that diversity, collaboration, social components, and music accomplishment are key to community music activities for well-being promotion. Based on these findings we recommend the following in implementing and providing community music activities: (1) a wide range of musical opportunities to facilitate music engagements of participants; (2) collaboration with community stakeholders for planning and organization to recruit participants and ensure resources; (4) collaborative music activities that may involve various specialists from different areas and participants; (5) employment of activity components to facilitate social engagement; (6) employment of activity and music component that facilitate the active engagement of participants; and (7) employment of components that can reinforce the sense of musical accomplishment through facilitating motivation.

### 4.5. Limitations

Some limitations to our study exist. First, despite attempts to use a comprehensive search strategy to identify publications in gray literature, inclusion of other terms and electronic databases may have yielded additional publications. The review also only included publications in English. Therefore, additional relevant studies might have been missed. Second, inclusion of reports on existing, ongoing community music activities and planned community music activities, with characteristics that are closer to music interventions, may have influenced the results. Although both types, in a broader sense, employ community music activities, ongoing community music activities that typically occur in naturalistic settings may not have been suitable for a strict review summary, which differs from intervention studies. We also did not distinguish between community music therapy and community music activities that may have different approaches in terms of activity goals and process. Notwithstanding such limitations, this review is the first to investigate the research characteristics and implementation and music strategies in community music activities supporting well-being, and may provide comprehensive and practical information for planning future research and practice.

## 5. Conclusions

This scoping review provides comprehensive information about current research and practice in community music activities that address well-being. Findings of the review confirmed the wide variety and diversity of community music activities, as suggested in previous research. Distinctive from other music-based intervention studies, community music activities showed a wide range of population groups, from non-clinical to clinical, and younger to older individuals. The findings also showed that studies often considered groups that were formed through participation in community facilities connected by socio-economic factors, location, and health issues. This occurs because of the unique attributes of community music activities, which support well-being across various well-being domains. Thus, individuals can benefit through participation that meets their own needs but is also a social act that reflects the larger community. This review suggests that collaboration with individuals, such as volunteers and professionals from other disciplines, as well as community and government institutions are essential to implement community music activities for well-being. Finally, the results of studies that examine well-being benefits imply that a core aspect of community music activities that facilitate well-being is the act of ‘socially engaged music making’. 

## Figures and Tables

**Figure 1 ijerph-20-02606-f001:**
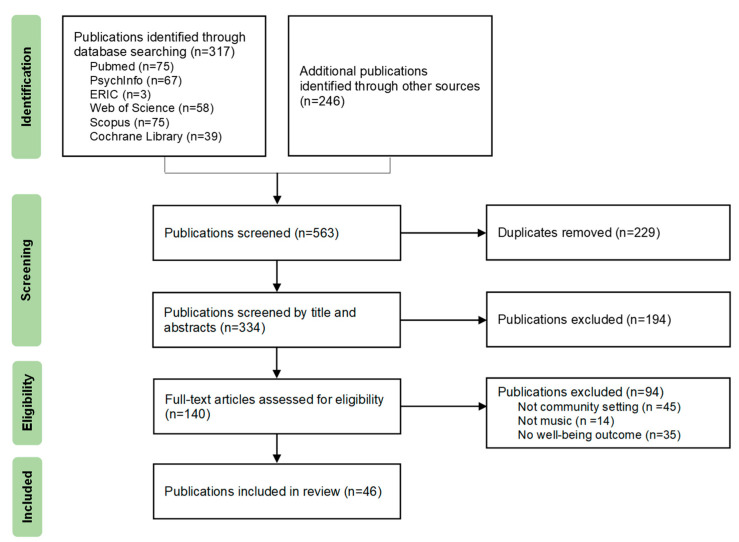
Flow diagram of study selection.

**Table 1 ijerph-20-02606-t001:** Inclusion and exclusion criteria for literature search.

Item	Inclusion Criteria	Exclusion Criteria
Participants	Participants of all age and condition affiliated in a “community”, defined as a group of people with diverse characteristics who are linked by social ties, share common perspectives, and engage in joint action in geographical locations or settings	Participants recruited in clinical settings
Concept	Any studies that address music intervention or activities as the main intervention undertaken in community settings	Music intervention or activities undertaken in clinical context; arts-based activities
Context	Studies that include outcomes of well-being, including mental health or quality of life	
Evidence sources	Studies published in English with quantitative and qualitative methodologies accounting for music intervention or activity program	Case studies

## Data Availability

Not applicable.

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
