# Peer review of "Implementation and Strategies of Community Music Activities for Well-Being: A Scoping Review of the Literature"

_ijerph, 2023, doi:10.3390/ijerph20032606_

Round 1

Reviewer 1 Report

The work addresses a topic already studied. However, the topic remains interesting.

The part that needs to be improved is the methodology, where it is necessary to put the "MeSH" words used, their combination and the databases where they were searched.

It is also very important to justify the reason for the absence of research in the gray literature. As this is a common theme in clinical practice, there must be guidelines that may be important in the study. On the other hand, linguistic issues should also be discussed. Example: the use of music in English, Portuguese, or Spanish can have different impacts.

Author Response

The work addresses a topic already studied. However, the topic remains interesting.

Response: Thank you very much for your comments that helped us improve our manuscript. We tried our best to address all comments.

The part that needs to be improved is the methodology, where it is necessary to put the "MeSH" words used, their combination and the databases where they were searched

Response: We added the exact MeSH search terms used in the search strategy. However, during the search process, we have iteratively searched databases in numerous combinations of search terms, in accordance with the aim of scoping review to identify as many publications as possible. Therefore, we stated the search strategy using Boolean operators and all databases.  

It is also very important to justify the reason for the absence of research in the gray literature. As this is a common theme in clinical practice, there must be guidelines that may be important in the study.

Response: We agree with your comment. For this reason, we tried to search the grey literature by manually searching relevant journals and reference lists separate to electronic database searches as suggested by scoping review guidelines of Arksey & O’Malley(2005). We consider that the description of the process and limitations written in the original manuscript, might have been misleading, so we rephrased descriptions of our search process (page 3 line 97) and limitations.

On the other hand, linguistic issues should also be discussed. Example: the use of music in English, Portuguese, or Spanish can have different impacts.

Response: Additional to the original report on music strategies regarding cultural background of participants, we have provided results on studies that specified using their own language among songs (page 7 line 290).  

Reviewer 2 Report

Background: 

Community music festivals play a great role in enhancing community cohesion, promoting economic health and community well-being. Of late, community events including music festivals have been organized as major tourism products in many travel destinations. In this context, presenting a scoping review “to synthesize published literature pertaining to community music activities for well-being promotion and identify the key implementation characteristics and strategies to inform future practice and research” seems a welcome step in the field.

Methodology applied in searching existing literature sounds in-depth and systematic. However, there are some issues and omissions in the manuscript which need to be addressed.

Omissions/Issues to be addressed:

1. Providing a standard definition of Community in Introduction somewhere before moving to the Materials and Methods is required so that readers develop a perspective what you are going to discuss. Therefore, authors are suggested to incorporate a definition of community reviewing the suitable literature, article/s in the field.

2. Economic well-being/benefits through community music festivals is missing in page 2 (lines 48-49)

Especially, research show benefits of community music activities across all emotional, psychological, and social dimensions of well-being [14,15].

Many communities organize events and festivals for economic gains; however, it is surprising that comprehensive literature review could not catch any aspect relating to economic gain/revival through organizing musical event.

Economic outcome/benefits component is totally missing in both qualitative and quantitative outcomes. Hence, it is recommended that this component (economic benefit) is properly included in study outcomes and discussions by revisiting the literature or redoing the search including community festivals/events and economic benefits/wellbeing. Authors are advised to review and reflect from more articles including the article below which discusses four dimensions of community sustainability.

Dangi, T. B., & Jamal, T. (2016). An integrated approach to “sustainable community-based tourism”. Sustainability8(5), 475.

3. Grammar and sentence structure should be reviewed at many places including:

15 studies 194 (33%) included socially (page 5 line 194). It should be fifteen in place of (15) at the start of the sentence.

Recommendations:

The manuscript is timely and worth publishing once it addresses the issues/omissions outlined by incorporating the recommendations made for improvement.

Author Response

Background:

Community music festivals play a great role in enhancing community cohesion, promoting economic health and community well-being. Of late, community events including music festivals have been organized as major tourism products in many travel destinations. In this context, presenting a scoping review “to synthesize published literature pertaining to community music activities for well-being promotion and identify the key implementation characteristics and strategies to inform future practice and research” seems a welcome step in the field.

Methodology applied in searching existing literature sounds in-depth and systematic. However, there are some issues and omissions in the manuscript which need to be addressed.

Response: We appreciate your valuable comments. We have carefully considered to comments and tried our best to address all points to improve our manuscript.  

Omissions/Issues to be addressed:

  1. Providing a standard definition of Community in Introduction somewhere before moving to the Materials and Methods is required so that readers develop a perspective what you are going to discuss. Therefore, authors are suggested to incorporate a definition of community reviewing the suitable literature, article/s in the field.

Response: We added a standard definition of community and conceptions of community music in the introduction (page 1 line 29) as suggested.

  1. Economic well-being/benefits through community music festivals is missing in page 2 (lines 48-49)

Especially, research show benefits of community music activities across all emotional, psychological, and social dimensions of well-being [14,15].

Many communities organize events and festivals for economic gains; however, it is surprising that comprehensive literature review could not catch any aspect relating to economic gain/revival through organizing musical event.

Economic outcome/benefits component is totally missing in both qualitative and quantitative outcomes. Hence, it is recommended that this component (economic benefit) is properly included in study outcomes and discussions by revisiting the literature or redoing the search including community festivals/events and economic benefits/wellbeing. Authors are advised to review and reflect from more articles including the article below which discusses four dimensions of community sustainability.

Dangi, T. B., & Jamal, T. (2016). An integrated approach to “sustainable community-based tourism”. Sustainability, 8(5), 475.

Response: We appreciate your kind reminder. We have also carefully reviewed the suggested article. As our research focus was well-being benefits among participants of community music activities, our outcome only considered well-being aspects that can be defined as positive state experienced by individual and societies (WHO, 2021). Also, sustainable community-based tourism from the suggested article focuses on the development and management of tourism activities, which we don’t consider as our main research focus, as community-based music activities included in our study aimed to improve well-being of community members. However, we think that it may be valuable to investigate this theme in future research.

  1. Grammar and sentence structure should be reviewed at many places including:

15 studies 194 (33%) included socially (page 5 line 194). It should be fifteen in place of (15) at the start of the sentence.

Response: Thank you for your comment. We went through the entire manuscript to eliminate grammatical mistakes.

Recommendations:

The manuscript is timely and worth publishing once it addresses the issues/omissions outlined by incorporating the recommendations made for improvement.

Response: Thank you very much for agreeing with us to the intention of this manuscript and helping us to improve our manuscript.

Round 2

Reviewer 1 Report

I understand the issue of interactive research.

However, in the methodology, it is very important to provide information that can replicate the research, as this is the only way to assess the rigor of the investigation.

Therefore, I suggested the inclusion of an attached table, where the different search phrases used, sequentially, and the respective databases could be placed.

The idea is that anyone, anywhere can enter the keywords used and get the same results.

Author Response

I understand the issue of interactive research.

However, in the methodology, it is very important to provide information that can replicate the research, as this is the only way to assess the rigor of the investigation.

Therefore, I suggested the inclusion of an attached table, where the different search phrases used, sequentially, and the respective databases could be placed.

The idea is that anyone, anywhere can enter the keywords used and get the same results.

Response: Thank you very much for pointing this out. We added the search date in text and the search strings of respective databases as a supplementary table, as attached.

Reviewer 2 Report

The manuscript seems well-revised though it could not include the aspect of economic gains/benefits through organizing community festivals (as suggested). Despite that shortcoming, it seems worth publishing. 

Author Response

The manuscript seems well-revised though it could not include the aspect of economic gains/benefits through organizing community festivals (as suggested). Despite that shortcoming, it seems worth publishing.

Response: We appreciate for your precious time in reviewing our paper that helped us improve our manuscript.